# Baicalin Protects Broilers against Avian Coronavirus Infection via Regulating Respiratory Tract Microbiota and Amino Acid Metabolism

**DOI:** 10.3390/ijms25042109

**Published:** 2024-02-09

**Authors:** Haipeng Feng, Jingyan Zhang, Xuezhi Wang, Zhiting Guo, Lei Wang, Kang Zhang, Jianxi Li

**Affiliations:** 1Engineering and Technology Research Center of Traditional Chinese Veterinary Medicine of Gansu Province, Lanzhou Institute of Husbandry and Pharmaceutical Sciences, Chinese Academy of Agricultural Sciences, Lanzhou 730050, China; fenghaipeng@caas.cn (H.F.); zhangjingyan@caas.cn (J.Z.); wanglei@caas.cn (L.W.); guozhiting@caas.cn (Z.G.); 2Lanzhou Veterinary Research Institute, Chinese Academy of Agricultural Sciences, Lanzhou 730050, China; wangxuezhi@caas.cn

**Keywords:** baicalin, IBV, respiratory tract microbiota, amino acid metabolism, broilers, anti-viral effect

## Abstract

An increasing amount of evidence indicates that Baicalin (Bai, a natural glycosyloxyflavone compound) exhibits an antiviral effect against avian viruses. However, it remains unclear if the antiviral effect of Bai against infectious bronchitis virus (IBV) is exerted indirectly by modulating respiratory tract microbiota and/or their metabolites. In this study, we investigated the protection efficacy of Bai in protecting cell cultures and broilers from IBV infection and assessed modulation of respiratory tract microbiota and metabolites during infection. Bai was administered orally to broilers by being mixed in with drinking water for seven days. Ultimately, broilers were challenged with live IBV. The results showed that Bai treatment reduced respiratory tract symptoms, improved weight gain, slowed histopathological damage, reduced virus loads and decreased pro-inflammation cytokines production. Western blot analysis demonstrated that Bai treatment significantly inhibited Toll-like receptor 7 (TLR7), myeloid differentiation factor 88 (MyD88) and nuclear factor kappa-B (NF-κB) expression both in cell culture and cells of the trachea. Bai treatment reversed respiratory tract microbiota dysbiosis, as shown by 16S rDNA sequencing in the group of broilers inoculated with IBV. Indeed, we observed a decrease in *Proteobacteria* abundance and an increase in *Firmicutes* abundance. Metabolomics results suggest that the pentose phosphate pathway, amino acid and nicotinamide metabolism are linked to the protection conferred by Bai against IBV infection. In conclusion, these results indicated that further assessment of anti-IBV strategies based on Bai would likely result in the development of antiviral molecule(s) which can be administered by being mixed with feed or water.

## 1. Introduction

The subfamily *Orthocoronavirinae* is classified within the family *Coronaviridae*, and contains four genera: *Alphacoronavirus*, *Betacoronavirus*, *Gammacoronavirus* and *Deltacoronavirus* [1]. The avian infectious bronchitis virus (IBV) belongs to genus *Gammacoronavirus*, which mainly infects birds, including turkeys [2]. The IBV genome is composed of a non-segmented single-stranded RNA of positive polarity containing six genes that are flanked by 5′ and 3′ non coding regions (total length ~27.6 kb). The genome accumulates mutations due to the error-prone nature of the polymerase and is subject to recombination due template switching in the S1 gene sequence [3]. IBV replicates primarily in the respiratory epithelium of chickens, resulting in damage to the tracheal cilia, which becomes more susceptible to secondary bacterial infection [4]. Consequently, IBV infection represents a major threat to broilers’ breeding, resulting in huge economic losses for the global poultry industry. Currently, vaccination is the main measure for preventing and controlling infectious bronchitis (IB), but recombination and mutation of IBV may compromise the effectiveness of vaccination. Therefore, broad-spectrum antiviral strategies based on molecules that could be administered through mixing with feed or water are likely to be an effective complement that would help control or prevent IB.

*Scutellaria baicalensis* Georgi is a medicinal plant that has been long used in traditional medicine in China to treat diseases, mainly due to its bitter taste and cold nature. In traditional Chinese medicine, *Scutellaria baicalensis* Georgi is considered to possess detoxifying and heat-clearing effects.

Baicalin (Bai) is a glycosyloxyflavone that is extracted from the root of *Scutellaria baicalensis* Georgi. An increasing amount of evidence strongly suggests that Bai can treat avian respiratory tract diseases caused by viruses such as the *Avian influenza virus* (AIV), *Newcastle disease virus* (NDV), *Avian leukemia virus* (ALV) or the bacterium *Mycoplasma gallisepticum* (MG) [5,6,7,8]. An anti-inflammatory effect of Bai is attributed to the inhibition of the Toll-like receptor 4 (TLR4)/nuclear factor kappa B (NF-κB) pathway. Bai also attenuates MG-induced inflammation by inhibiting the TLR2/NF-κB pathway [9,10]. The anti-inflammatory properties of Bai and its low toxicity make it an interesting compound for the potential treatment of viral respiratory tract diseases [11,12]. Our previous findings indicate that Bai significantly inhibited the replication of the IBV Beaudette strain in Vero cells [13]. Despite the positive effect of Bai on respiratory tract microbiota, its potential antiviral effect has not been assessed in chickens. A published study revealed that Bai promoted microbial diversity and the ratio of *Firmicutes*/*Bacteroidetes* in MG infected broilers [6]. 

The respiratory microbiome plays a pivotal role in the pathogenesis of viral infection. Respiratory tract microbiota dysbiosis disrupts pulmonary immune homeostasis and predisposes to lung disease [14]. Viral infection can disrupt the delicate balance between local defences and bacterial growth in the respiratory tract. Several mechanisms have been suggested which predominantly refer to alteration of the diversity and abundance of lung floras [15]. Viral infection alters the relative abundance of bacterial taxa in the respiratory tract, including *Pseudomonas*, *Corynebacterium*, and *Streptococcus* [16]. Traditional Chinese medicine plays a crucial role in modulating respiratory tract flora to combat viral infections. For instance, Mahuang Fuzi Xixin decoction has been reported to mitigate damage to the epithelial barrier of lung and nasal mucosa by regulating lung microbiota and rectifying plasma metabolism imbalance [17]. Therefore, the respiratory tract microbiota, together with its associated metabolites, may play an important role in mitigating the duration and perhaps the severity of respiratory virus infection.

In the current study, we assessed the potential anti-inflammatory effects of Bai in both chicken cell culture and broilers. Our results suggest that Bai exerted an anti-inflammatory effect by modulating the TLR7/MyD88/NF-κB pathway. The underlying mechanisms of protection from IBV infection appear to involve regulation of amino acid metabolism and associated microbiota in the respiratory tract. In summary, the potential of Bai as a complement to vaccination for treating tracheal injury induced by IBV infection is promising. 

## 2. Results

### 2.1. Assessment of Anti-IBV Effect of Bai in CEK Cell

The cytotoxicity of various concentrations of Bai was assessed in CEK cells. Cell viability was greater than 92% when the concentration of Bai was not greater than 125 μg/mL. Cytotoxicity increased with increasing Bai concentration. Cell viability was less than 63% when the concentration of Bai was greater than 125 μg/mL, as shown in Appendix A. IFA results showed that the expression levels of IBV N protein were reduced significantly when IBV-infected cells were treated with Bai for 24 and 36 h (Appendix A). RNA copy numbers of IBV in CEK cells treated with 125 μg/mL Bai for 36 h were significantly decreased (*p* < 0.05) compared to untreated cells infected with IBV (Appendix A).

### 2.2. Bai Inhibited TLR7/MyD88/NF-κB Signalling Pathway and its Downstream Inflammatory Factors in CEK Cell

The levels of IL-1β, IL-6 and TNF-α in CEK cells are shown in Appendix A. These results show that the levels of IL-6 and TNF-α mRNAs in cells treated with 125 μg/mL of Bai were significantly decreased at 24 hpi (*p* < 0.05) compared with infected but untreated cells. However, there was no significant difference at 36 hpi (*p* > 0.05). Bai treatment had no influence on the levels of IL-1β (*p* > 0.05) mRNA. Bai treatment of infected CEK cells also reduced the levels of expressed protein for IL-1β, IL-6 and TNF-α at 24 and 36 hpi compared to infected but untreated cells, and also when compared to uninfected CEK cells treated with Bai only (125 μg/mL), as shown in Figure 1. The levels of mRNA and protein expression for TLR7 and MyD88 at 36 hpi were significantly inhibited in CEK cells treated with 125 μg/mL of Bai, particularly compared to IBV-infected but untreated cells. No significant difference was observed at 24 hpi (*p* > 0.05), as shown in Appendix A and Figure 2A,B. Furthermore, Bai treatment (125 μg/mL) specifically inhibited the phosphorylation level of NF-κB p65 in infected and treated CEK cells compared to infected but untreated cells; however, we did not observe significant differences in levels of mRNA expression (*p* > 0.05) (Appendix A and Figure 2D).

### 2.3. Observation of Clinical Signs and Follow up of the Changes in Body Weight

Throughout the infection period (days at 13–18), the weight gain in IBV-infected broilers was notably slower than the control group. Broilers exhibited a range of clinical signs upon being challenged with IBV, including ruffled feathers, neurological signs, clustering and gathering behaviour, and wing drooping. Bai treated infected broilers showed a significantly higher body weight (Table 1). At the same time, the severity of clinical symptoms of broilers in the Bai-treated/IBV-infected group was reduced compared to the untreated IBV-infected group (Appendix A), and there were no clinical symptoms in the control uninfected and untreated group during the whole experiment.

### 2.4. Histopathology and Scanning Electron Microscopy Analysis

We examined the histopathological changes in the trachea, focussing mainly on the integrity of the respiratory epithelium and presence/absence of inflammatory cell infiltration in tracheal mucosa. In untreated IBV-infected broilers, the cilia and epithelial cells exfoliated, with slight haemorrhages, and we observed a large number of monocytes and lymphocytes infiltrating the mucosal layer. Infected broilers that have been treated with Bai showed smaller numbers of inflammatory cells infiltrating the mucosa, as well as an almost intact structure of epithelial cilia (minor damage, Appendix A). In untreated IBV-infected broilers, the lungs showed severe haemorrhaging in the parabronchial and vascular lumens. IBV infection did not cause severe pathological change in Bai-treated broilers, and only minor haemorrhaging in the parabronchial lumen was observed, as shown in Appendix A. These results indicated that Bai significantly protects the structure of the tracheal mucosa.

The changes to ultrastructural morphology in the trachea were assessed by SEM. In the uninfected/untreated control group, tracheal ultrastructures were normal, showing well-organised ciliary arrangements. Brush cells were dispersed throughout. The tight junctions between cells were well observed with intact cell membranes. Additionally, ciliated cells were evenly distributed (Figure 3A,B). In the IBV-infected group of broilers, the tracheal cilia were damaged/exfoliated, displaying shrunken brush cells and damaged cell membranes. Large surfaces had a convex appearance and showed degenerated cilia (Figure 3C). The observed damage in infected untreated broilers was absent in broilers which had been pre-treated with Bai, as shown in Figure 3D.

### 2.5. Anti-IBV Antibody Level in Serum

Specific anti-IBV antibody levels in serum were detected by the ELISA kit. The level of anti-IBV antibody in the untreated/IBV-infected group remained relatively low level compared to the control group during the entire experiment. Compared with the untreated/IBV-infected group, the level of antibodies gradually increased in the Bai-treated/IBV-infected group between day 0 and day 5 post-infection, and this increase was found to be statistically significant on days 3 and 5 post-infection, as shown in Appendix A.

### 2.6. Analysis of IBV and G3BP1 in Trachea

Virus loads in tracheas showed a similar trend to that featured in the lung. Briefly, viral loads in the trachea reached their highest levels at 3 dpi before gradually declining by 5 dpi. Viral loads in trachea were significantly lower in Bai-treated/IBV-infected group compared to the untreated/IBV-infected over the course of the experiment. However, overall viral loads in lungs were lower compared to that in the trachea, as shown in Appendix A. Furthermore, IHC result showed that the IBV N protein was predominantly found in the tracheal mucosa surface (Figure 4A). The peak of IBV detection in the tracheal mucosal surface was reached on day 3 post-infection in the untreated/IBV-infected group (Figure 4C). Compared to the untreated/IBV-infected group, the level of G3BP1 protein (an antiviral protein) in the Bai-treated/IBV-infected group was increased and reached its highest at 3 dpi. In addition, IBV infection reduced levels of mRNA and protein expression of G3BP1 in the cells of the tracheal surface, as shown in Figure 5B,D and Figure 6D.

### 2.7. Effect of Bai on the TLR7/MyD88/NF-κB Signalling Pathway and Its Downstream Signal Protein in Trachea

To investigate the potential impact of Bai on the inflammatory response, levels of pro-inflammatory cytokines in the trachea were assessed using RT-qPCR and WB analyses. IBV infection significantly increased the mRNA expression of pro-inflammatory cytokines (IL-1β, TNF-α, IL-6) in trachea compared with the control group (Figure 5A–C). As expected, the WB result also confirmed that IBV infection upregulated the levels of protein expression for these cytokines (Figure 6F–H). However, Bai treatment significantly reduced levels of the corresponding mRNAs, which were nearly comparable to levels of the same mRNAs in the control group. Furthermore, Bai treatment downregulated levels of protein expression for these cytokines. These results indicated that Bai treatment reduced the level of the pro-inflammatory cytokines (IL-1β, TNF-α, IL-6) in the IBV group; therefore, the potential protective effect of Bai against IBV may be attributed to its anti-inflammatory effect. 

We further assessed levels of the mRNA and proteins for the TLR7MyD88/NF-κB pathway. Gene expression of TLR7, MyD88 (*p* < 0.05) and downstream NF-κB p65 (*p* < 0.01) were significantly increased in the IBV group (Figure 5E,G,H). Conversely, Bai treatment led to a significant downregulation as compared to the untreated/IBV-infected group. This was also confirmed by WB, which showed that levels of TLR7 and MyD88, as well as the level of phosphorylation of NF-κB p65, were significantly decreased in the Bai-treated/IBV-infected as compared to the untreated/IBV-infected group (Figure 6A–C). Overall, our results indicated that Bai alleviated inflammation caused by IBV via an inhibiting TLR7/MyD88/NF-κB signal pathway.

### 2.8. Bai Corrected the Dysbiosis of Respiratory Flora Caused by IBV Infection

#### 2.8.1. *Alpha* and *Beta* Diversity Analysis

It is widely known that respiratory dysbiosis is closely associated with the release of pro-inflammatory cytokines and favoured secondary infection by bacteria. In our study, we calculated α diversity index to evaluate species richness and diversity, including Shannon, Simpson and Chao1 indices. As shown in Appendix A, the Shannon and Simpson index showed no significant change in all groups, while the Chao1 index was increased in the Bai-treated/IBV-infected, as compared to untreated/IBV-infected group (Appendix A). The *beta* diversity was analysed based on UniFrac’s weighted PCoA, and the results showed a relative separation between untreated/IBV-infected and Bai-treated/ IBV-infected groups, indicating that Bai improved the imbalance in respiratory microbiota, as shown in Figure 7C.

#### 2.8.2. The Regulatory Effect of Bai on the Abundance of Respiratory Microbiota

We analysed the changes in respiratory microbiota at the phylum, family and genus levels, as shown in Figure 8. In our study, at the phylum level, IBV infection resulted in a significant increase in the relative abundance of *Proteobacteria*, while there was a notable decrease in *Proteobacteria* and an increased abundance of *Firmicutes* in the Bai-treated/IBV-infected group. In addition, Bai treatment decreased the abundance of *Herminiimonas* at the genus level. The results showed that Bai treatment helped in improving microbiota composition in the respiratory tract. To further explore the differences in microbial communities among different groups, we estimated the impact of different abundances based on the LEfSe linear discriminant analysis. The LDA analysis revealed significant variations in dominant bacterial composition among all groups. *Firmicutes*, *Bacilli* and *Mycoplasmatales* were present in the untreated/uninfected control group, *Bacteroidota*, *Enterobacterales*, *Pasteurellaceae*, etc., were present in the untreated/IBV-infected group. In the Bai-treated/IBV-infected group, it was *Proteobacteria*, *Gamma-proteobacteria*, *Oxalobacteraceae*, etc., as shown in Figure 7A,B. These results indicated that IBV infection altered respiratory microbiota composition and population structure, and Bai protected from tracheal injury by improving regulation with the respiratory microbiota.

### 2.9. Bai Changed Metabolomics Changes in Trachea

The principal component analysis (PCA) result showed that there was a significant difference, with good reproducibility, within all groups. In our study, the PCA analyses of the QC samples revealed a significant degree of differentiation in the metabolic profiles and excellent reproducibility, indicating that the data was of good quality and the instrument was stable (Figure 9A).

We used partial least squares discriminant analysis (PLS−DA) to analyse the differential metabolites among three groups. The result indicated that samples in all groups were obviously separated in both positive and negative ion modes (Figure 9B,C). The clustering heat map of differential metabolites showed that IBV infection induced a change in various components, such as Kinetin, 3-Hydorxy-3-methylglutaric acid, N-Acetylserine, L-Hydroxyproline, Deoxyguanosine, Thymidine, Deoxyuridine, m−Coumaric acid, Hydroxyphenyllactic acid, 4-Hydroxyphenylpyruvic acid, Phenylpyruvic acid, 2-Oxo-4-methylthiobutanoic acid, Phenylpyruvic acid and *alpha*-Ketoca proic acid. IBV also decreased the level of L-Pyroglutamic acid, N-Acetylglycine, Deoxyribose 1-phosphate and Maleamic acid in tracheas. As shown in Figure 9D,E, Bai significantly downregulated the level of N-Formyl-L-methionin, 3,4-Dihydroxyhydrocin namic acid, Acetovanillone, p-Cresol glucuronide, Oxypurinol, Quinolinic acid, N-Acetyl-L-methionine and Quinic acid, while it also regulated D-Ribose levels.

The KEGG analysis showed that IBV infection induced changes in metabolic pathways, such as Pyrimidine metabolism, 2-Oxocarboxylic acid metabolism, ABC transporters, Pentose phosphate pathway, Tyrosine metabolism, Phenylalanine metabolism, Phenylalanine, tyrosine and tryptophan biosynthesis, D-Amino acid metabolism, Arachidonic acid, Ubiquinone and other terpenoid-quinone biosynthesis metabolism, Biosynthesis of amino acids, Purine metabolism, Cysteine and methionine metabolism and the Biosynthesis of cofactors. Bai downregulated Tyrosine metabolism, Tryptophan metabolism, Cysteine and methionine metabolism, Biosynthesis of cofactors, Nicotinate and nicotinamide metabolism, *beta*-Alanine metabolism pathways, while it also upregulated ABC transporters and Pentose phosphate pathways (Figure 10A,B).

### 2.10. Correlation Analysis of Differential Metabolites and Microbiota Composition

The potential association between genera of microbiota and metabolites in the trachea was determined using a Spearman correlation analysis. As shown in Figure 10C, there was a significantly positive correlation between *Pseudomonas* and Quinolinic acid, N-Formyl-L-methionine, Herminiimonas, Quinolinic acid, *Bacteroides* and 3, 4-Dihydroxyhydrocinnamic acid at the genus level. Meanwhile, *Blautia* exhibited a negative correlation with N-Formyl-L-methionine. At the phylum level, *Firmicutes* was negatively correlated with Quinolinic acid, N-Formyl-L-methionine, Allopurinol-1-ribonucleoside and Acetovanillone, Kynurenic acid, Verrucomicrobiota, and it was negatively correlated with N-Formyl-L-methionine and N-Acetyl-L-methionine. *Chloroflexi* was negatively correlated with p-Cresol glucuronide and 3-Methoxy-4-hydroxypheny-lethyleneglycol sulfate. *Bacteroidota* and 3,4-Dihydroxyhydrocinnamic acid along with Campylobacterota and 3-Methoxy-4-hydroxyphenylethyleneglycol sulfate were positively correlated (Figure 10D), suggesting that *Pseudomonas*, *Mycoplasma*, *Herminiimonas*, *Bacteroides* and *Blautia* had pivotal effects on the protective effect afforded by Bai against IBV infection.

## 3. Discussion

IBV, as a causative agent of viral respiratory diseases, has become endemic throughout the poultry industry since being first documented in the United States in 1931 [18]. It infects chickens of all ages; however, younger chicks are particularly vulnerable and show more severe symptoms, thereby resulting in substantial economic losses for the poultry industry. The lack of cross-protection among different serotype strains have hampered the implementation of effective measures for controlling IB [19], and so it is therefore crucial and urgent to develop an effective antiviral strategy to counter IBV. Bai is a major biological active ingredient extracted from *Scutellaria baicalensis* Georgi and exhibits a range of pharmacological effects, including potential antiviral effects. 

We explored the inhibitory effect of Bai against IBV M41 in vivo and in vitro, assessing the mechanistic basis. We found that Bai treatment reduces cilia exfoliation and the number of brush cells, protects ciliary structure integrity, enhances the level of anti-IBV antibodies and reduces the level of viral RNA copies in treated cells. Additionally, when IBV-infected broilers were administered with Bai, the abundance of *Proteobacteria* was decreased while the abundance of *Firmicutes* was increased, and metabolites were enriched in the pentose phosphate pathway, amino acid and nicotinamide metabolism. The evidence presented in this study suggests that the protection afforded by Bai against IBV infection was associated with the modulation of the dysbiosis of respiratory microbiota and metabolites composition. Meanwhile, the expression levels of inflammatory cytokines and TLR7/MyD88/NF-κB pathway indicates that Bai reduced the inflammatory injuries in IBV-infected broilers.

Our previous study showed that IBV replication in CEK cells peaked at 36 hpi [20]. In the current study, we found that IBV RNA copies were decreased at 36 hpi in IBV-infected cells after being treated with Bai. Our results were consistent with previous studies, which confirmed that Bai exerted anti-IBV effects [21,22]. Despite achieving satisfactory results in vitro, some drugs showed poor efficacy in animal experimentation. We therefore assessed whether Bai could prevent IBV infection in broilers.

IBV-infected broilers showed significant weight loss, severe respiratory symptoms, ciliary structure injuries, extensive inflammatory cell infiltration and higher pathological scores. The decrease in body weight is a primary manifestation of IBV infection, and it significantly contributes to substantial economic losses incurred for poultry farmers [23]. In our study, Bai treatment reduced clinical signs and protected against decreased body weight. Our findings are in agreement with previous studies involving the *influenza virus* [24]. Therefore, we speculated that Bai may help prevent economic losses caused by IBV. Histopathology and scanning electron microscopy observation showed that IBV infection damaged tracheal ciliary structures, with a large number of epithelial cells infiltrating the mucosa and exfoliated ciliary cells, which altogether increased susceptibility to bacterial infections. Bai pre-treatment prevented severe damage caused by IBV infection to tracheal ciliary structure and significantly reduced exfoliation in epithelial cells. This is similar to previous findings about Bai reducing MG-induced lung injuries and LPS-induced liver injuries [6,12]. The mechanical barrier of tracheal mucosa is formed by the intercellular junctions between tracheal epithelial and adjacent cells, which constitutes a vital component of tracheal mucosal immunity and effectively hinders macromolecule passage. This mechanism inhibits bacterial and viral diffusion into the body through tracheal mucosa [25]. Taken together, these results indicated that Bai effectively preserved tracheal structural integrity, thereby contributing to the protection of broilers against IBV invasion.

The pathogenesis of IBV is characterised by elevated viral RNA copies, enhanced inflammatory response, and delayed initiation of innate and adaptive immune responses. The previous research demonstrated that IBV-induced tissue damage not only contributes to lung injury, but also to an excessive production of inflammatory cytokines which can exacerbate tissue damage and impair gas exchange. Ultimately, this leads to tracheal embolism and respiratory distress [26]. Our results found that Bai not only decreased the levels of IL-1β, IL-6 and TNF-α, but that it also upregulated the expression of G3BP1 in the tracheal cells. An earlier study showed that the degree of pathological changes is positively correlated with the release of inflammatory factors during IBV infection [27]. Bai possesses anti-inflammatory properties, especially alleviating inflammatory responses induced by viral and bacterial infections [11,28]. The NF-κB pathway serves as a pivotal inflammatory cascade regulator, while the viral-induced NF-κB pathway activation significantly contributes to the generation of pro-inflammatory cytokines [29]. TLRs also play crucial regulatory roles in inflammatory response. Various IBV strain infections have induced the activation of TLR7/MyD88 pathways, thereby exacerbating the severity of lesions [30]. A large number of studies showed that natural products suppressed the secretion of inflammatory cytokines via TLRs-mediated NF-κB pathway [31,32]. Hence, the inhibition of TLRs-mediated NF-κB inflammatory pathways may reduce IBV-induced tissue damage. Our results demonstrated that Bai reduced the expression of TLR7/MyD88 and inhibited the phosphorylation of NF-κB p65. These findings proved that the anti-inflammatory effect of Bai in broilers may be related to TLR7/MyD88/NF-κB signal pathway, which was also exhibited via in vitro experimentation.

The respiratory microbiome may influence the host immune response and modify the susceptibility to viruses. A previous report indicated that MG infection altered respiratory microbiota composition in birds [33]. The imbalance of *Proteobacteria* is often associated with micro-ecological disorders, and there is a view that the increase in *Proteobacteria* is considered as a factor for disease diagnosis [34]. Previous studies have demonstrated that the abundance of *Proteobacteria* was markedly increased in mice with asthma and some human inflammatory diseases [35,36]. Our study confirmed the view that the increase in *Proteobacteria* in the respiratory tract may be related to the degree of tissue damage induced by IBV infection, as we observed a significant alteration in both abundance and composition of respiratory microbiota due to IBV infection, potentially leading to an increase in the abundance in *Proteobacteria*. Other study also indicated that exogenous exposure (high concentration of ammonia) disturbed the ecological balance of broiler’s respiratory flora and increased the proportion of *Proteobacteria* [37]. Interestingly, Bai treatment significantly decreased *Proteobacteria* abundance. *Firmicutes*, *Mycoplasma*, *Proteobacteria* are related to MG and AIV infection according to previous literature, which may be related to a variety of inflammatory diseases and aggravate the inflammatory injury degree [38,39]. *Alpha* diversity analysis showed no significant difference in flora abundance, while *beta* diversity analysis revealed significant differences in microbial communities among all groups. Therefore, this study demonstrated that specific alterations in respiratory microbiota are associated with the inflammation injuries induced by IBV infection.

The use of metabolomics was proven to be invaluable for investigating the pathogenesis of IBV infection. In our work, IBV infection significantly increased the level of various amino acids and decreased the level of L-Pyroglutamic acid, N-Acetylglycine, Deoxyribose 1-phosphate and Maleamic acid. Previous research found that these metabolites induced pulmonary surfactant system aberrations during *influenza virus* infection, which might play an important role in the etiology of respiratory failure and repair [40]. Bai significantly decreased eight metabolites, including Quinolinic acid involved in tryptophan metabolism, N-Formyl-L-methionin involved in glutamine metabolism, 3,4-Dihydroxyhydrocinnamic acid involved in Caffeic acid metabolism, p-Cresol glucuronide involved in Phenylalanine and complexine, Oxypurinol involved in Purine metabolism, Quinic acid involved in fat metabolism, as well as upregulated D-Ribose levels. These results indicated that Bai could restore the metabolic status in IBV-infected broilers to a normal level. The metabolism of amino acids in the body is intricately linked to the extent of inflammation damage. Bai can regulate arginine biosynthesis and alleviate inflammation damage induced by MDR *Pseudomonas aeruginosa* in acute pneumonia in rats [41]. The pathway enrichment analysis in our study revealed that Bai treatment has specific effects on the amino acid metabolism (Tyrosine, Phenylalanine, Tyrosine and Tryptophan) and steroid hormone biosynthesis. Previous studies also reported that *Gastrodia elata* polysaccharides and wine-processed radix scutellariae can change the amino acid metabolism in acute respiratory distress syndrome and inflammation-related diseases, such as tryptophan, phenylalanine, tyrosine, histidine and leucine metabolism [42,43]. Recently, faecal sample metabolomics analyses revealed a significant association between metabolic alterations and gut microbiota disorders in the context of inflammation-related disease progression, encompassing both viral and bacterial diseases. The correlation analysis was performed in this study to investigate the association between respiratory microbiotas at the genus and phylum levels with metabolites. The findings of our research demonstrated a robust correlation between these anti/pro-inflammatory bacteria (*Pseudomonas*, *Mycoplasma*, *Firmicutes*, *Bacteroides*) and alterations in amino acid metabolites. In summary, we found that Bai restored IBV induced amino acid metabolism dysregulation, which indicated that regulating amino acid metabolism is an important mechanism in Bai treating IBV.

## 4. Materials and Methods

### 4.1. Virus, Cell and Drugs

The IBV M41 strain was obtained from the China Institute of Veterinary Drug Control (Beijing, China) and passaged 10 times in specific pathogen-free (SPF) chicken embryos (10^−5.78^EID_50_/0.1 mL). The primary chicken embryonic kidney (CEK) cells were isolated from SPF chicken embryos in accordance with previously described protocols [44,45]. The CEK cells were incubated at 37 °C with 5% CO_2_. The Bai in this study was purchased from Shanghai yuanye Bio-Technology Co., Ltd. (Shanghai, China) and Shaanxi Senfu Natural Products Co., Ltd. (Xi’an, China). The purity of the Bai was over 97%, as determined by HPLC.

### 4.2. Animals

One-day-old unimmunised male white-feather broiler-type chickens were purchased from a local commercial broiler hatchery (the Jiuquan Wangmiao commercial hatchery, Jiuquan, China). Birds were reared in negative pressure isolators equipped with appropriate light and temperatures, and they had ad libitum access to food and water. All experimentation procedures were approved by the Animal Care and Use Committee, Lanzhou Institute of Husbandry and Pharmaceutical Science of Chinese Academy of Agricultural Science (permission number: SYXK (Gan) 2019-0002).

### 4.3. Cytotoxicity Assay

Cytotoxicity of Bai was assessed using a cck-8 assay in accordance with the manufacturer’s instruction (ZETA Life, Menlo Park, CA, USA). Briefly, CEK cells were seeded in 96-microwell plates at a density of 2 × 10^5^ per well. Cells were treated with various concentrations of Bai solution (with a range of concentration from 0 to 1000 µg/mL) and incubated at 37 °C. After 48 h, cells were washed twice with PBS and 10 μL cck-8 reagents were added. The absorbance of each well was measured using a Microplate Spectrophotometer at 450 nm.

### 4.4. Antiviral Activity

CEK cells were cultured at 37 °C for 24 h and washed twice with phosphate buffered saline (PBS). The virus suspension (100 EID_50_) was added and incubated for 2 h. Different concentrations of Bai (125 μg/mL, 100 μg/mL, 75 μg/mL) were then added to different wells. Total RNA and protein were extracted. Cells were also analysed by immunofluorescence. Briefly, CEK cells were washed three times with PBS. Treated cells were fixed with 4% paraformaldehyde, followed by permeabilisation with PBS containing 0.1% (*v*/*v*) Triton-X100 for 20 min at room temperature (RT). IBV N protein was detected using a mouse monoclonal antibody and a Goat anti-Mouse IgG AF488 secondary antibody. The sources of the antibodies are indicated in Table 2. Nuclei were stained with DAPI (4′6-diamidino-2-phenylindole, Solarbio, Beijing, China). The stained cells were observed using a fluorescence microscope (Olympus, Tokyo, Japan). The total fluorescence intensity was calculated with Image-J software version 1.8.0. The source of the IBV N antibody is shown in Table 1.

### 4.5. Animal Experimentation

A total of 140 chickens (7 days old) were randomly divided into 4 groups, including an uninfected control group, an IBV-infected group, a Bai (450 mg/kg BW per chick) treatment group and a group that was infected with IBV and treated with Bai group [46]. Bai was administered to birds by mixing with drinking water for 7 days. After this, the birds that were 14 days old were inoculated with 0.3 mL/bird of an IBV-M41 suspension (containing 10^4^ EID_50_ of IBV) through the ocular-nasal route. Clinical signs were recorded daily. Birds were euthanised on days 1, 3 and 5 post-infection (dpi). Tracheas were dissected out, and sections of them were fixed in neutral formalin solution for histopathological analysis. Portions of the tracheas collected 5 days post-infection were fixed in 2.5% glutaraldehyde for analysis by scanning electron microscopy (SEM). The rest of tracheas were placed in RNA later solution and stored at −80 °C for further analysis.

### 4.6. ELISA for Anti-IBV Antibodies

IBV-specific antibodies in serum were quantified by a highly sensitive commercial ELISA kit in accordance with the manufacturer’s instructions (Jiubang, Fujian, China).

### 4.7. Histopathological Detection and Immunohistochemistry Analysis (IHC)

Fixed tracheal tissues were embedded in paraffin. Sections of 5 μm of thickness were stained with haematoxylin and eosin. Slides were incubated in 3% hydrogen peroxide for 15 min at RT after being dewaxed and dehydrated. Slides were washed 3 times with Tris buffered saline (TBS), pH = 7.6 and incubated with the primary antibody (diluted 1:500) at 4 °C in a humidified chamber overnight after being blocked with 10% normal goat serum for 30 min at 37 °C. Further, tissue sections were incubated with a secondary antibody for 30 min at 37 °C and washed 3 times with TBS-Tween 20 (0.1% Tween 20) for 5 min. Slides were incubated with diaminobenzidine (DAB) for 5 min, washed with distilled water, counter-stained with haematoxylin for 3 min, dehydrated and observed under a light microscope (Olympus, Tokyo, Japan).

### 4.8. Scanning Electron Microscopy Analysis (SEM)

The tracheal tissues were fixed in 2.5% glutaraldehyde and washed 3 times with PBS, followed by treatment with 1% OsO_4_ for 1–2 h at RT. After 3 washes with PBS, and treatment with a graded series of ethanol and isoamyl acetates for 15 min, the samples were dried with a Critical Point Dryer and sputter-coated with gold for 30 s. Finally, the tissues were observed and had images taken with a scanning electron microscope (SEM, HITACHI, Tokyo, Japan).

### 4.9. Real-Time RT-qPCR

The total RNA from the CEK cells and tissues were extracted using TRIzol reagent (Takara, Dalian, China). The RNA was copied into cDNA using a PrimeScript RT reagent kit with gDNA Eraser (Takara, Dalian, China) in accordance with the manufacturer’s instructions. Gene expression analysis was performed using SYBR^®^Premix Ex Taq^TM^ II (Takara, Dalian, China) and a 7500 real-time PCR system (Applied Biosystems, Foster, CA, USA). The levels of mRNA expression were calculated using the 2^−ΔΔ^ threshold cycle (CT) method, and thre results were normalised to β-actin. The primer sequences are listed in Table 3.

### 4.10. Western Blot (WB) Analysis

The total proteins were extracted from CEK cells and tracheal tissue with a RIPA Lysis Buffer (Beyotime, Shanghai, China). Equivalent amounts of protein were separated by SDS-PAGE, followed by electroblotting onto polyvinylidene-difluoride membranes. The membranes were reacted with the primary antibody at 4 °C overnight after being blocked with QuickBlock blocking buffer, followed by washing three times with TBS-Tween 20. Then, membranes were incubated with a goat anti-mouse/rabbit IgG HRP secondary antibody for 1 h and washed with PBS-Tween 20 (0.1% Tween 20) three times. Finally, protein bands were visualised by chemiluminescence, and the grey value of the corresponding protein was analysed by Image-J software. The details of the antibodies used in this work are shown in Table 1.

### 4.11. Respiratory Microbial Structure Changes by 16S rDNA Analysis 

16S rDNA sequencing analysis was performed by Biotree Biological Technology Co., Ltd., Shanghai, China. In brief, bacterial genomic DNA samples were extracted from the broilers’ tracheas according to the E.Z.N.A^®^ Stool DNA Kit (D4015, Omega, Inc., Norwalk, CT, USA). The hypervariable region V3-V4 of the bacterial 16S rRNA gene was PCR amplified using the primer pairs 341F (5′-CCTACGGGNGGCWGCAG-3′) and 805R (5′-GACTACHVGGG TATCTAATCC-3′). The amplicons were cleaned up using AMPure XT beads (Beckman Coulter Genomics, Danvers, MA, USA). The amplicon pools were subjected to sequencing using an Illumina NovaSeq PE250 platform. The library’s depth and quantification were assessed using the Library Quantification Kit for Illumina (Kapa Biosciences, Woburn, MA, USA) and an Agilent 2100 Bioanalyzer (Agilent, Santa Clara, CA, USA). Specific sequences were converted into equivalent random sequences, and the resulting sequences’ alpha diversity and beta diversity were computed. The SILVA (version 132) classifier normalised the feature abundance based on the relative proportion of the gene sequences. The Shannon and Simpson indices in the samples were computed using QIIME2. The alpha diversity of the intestinal microbiota was calculated using the index of Shannon, Simpson, Chao and Operational Taxonomic Units (OTUs).

### 4.12. Analysis and Identification of Metabolites

Fractions of freeze-dried tracheas weighing 25 mg each were placed in Eppendorf tubes and extracted using an acetonitrile/methanol/water (2:2:1, *v*/*v*/*v*) mixture. The samples were homogenised for 4 min at 40 Hz in a rotating ball mill (JXFSTPRP-24, Shanghai Jingxin Technology Co., Ltd., Shanghai, China). This was followed by 3 successive cycles of sonication (ultrasonic generator PS-60AL, Shenzhen Redbond Electronics Co., Ltd., Shenzhen, China) for 5 min in ice water. The samples were subsequently incubated for 1 h at −40 °C and centrifuged (Heraeus Fresco 17, Thermo Fisher Scientific, Waltham, MA, USA) for 20 min at 5000 rpm. An internal standard (2-methylvaleric acid, 25 mg/L; Dr. Ehrenstorfer, Augsburg, Germany) was added to the supernatant together with 0.1 mL of 50% sulfuric acid (Sinopharm, Beijing, China). The mixture was incubated at −20 °C incubation for 30 min. This mixture was then transferred into a new glass vial for non-target metabolism analysis using the gas chromatography-mass spectrometry (GC-MS; GC2030-QP2020 NX, Shimadzu, Japan) technique. An HP-FFAP (30 m × 250 μm × 0.25 μm; J&W Scientific, Folsom, CA, USA) capillary column of the gas chromatography system (Agilent, Folsom, CA, USA) was filled with 1 µL supernatant. At an average flow rate of 1 mL/min, helium was employed as the transport gas. The initial operating temperature was kept at 80 °C for 1 min. It was then heated to 200 °C with ramp of 10 °C/min and maintained at 200 °C for 5 min. A further heating to 240 °C at 40 °C/min was introduced, and the operating temperature was kept at 240 °C for an additional min. Temperatures for the injection front, transfer line, quad, and ion source were set at 240 °C, 240 °C, 150 °C and 200 °C, respectively. The MS test was carried out with electron ionisation energy of −70 eV. The MS data with m/z spectra ranging from 33–150 were collected in Scan/SIM mode following a 3.5 min solvent delay.

### 4.13. Statistical Analysis

All data are presented as mean values ± SD. The comparison of groups was performed using one-way ANOVA. Repeated measurement data were analysed by two-way ANOVA. All statistical analyses were conducted using GraphPad Prism, version 7.0 (GraphPad, La Jolla, CA, USA) or R package. Differences were considered significant with *p* values less than 0.05. 

## 5. Conclusions

Our findings demonstrated the protection effect of Bai against IBV infection in broilers through restoring the respiratory microbiota dysbiosis and modulating amino acid metabolism. In addition, Bai attenuated tracheal injuries by inhibiting TLR7/MyD88/NF-κB signal pathway-mediated anti-inflammatory activities. Taken together, this research offers valuable empirical evidence and theoretical support for the potential application of Bai as an anti-IBV drug.

## Figures and Tables

**Figure 1 ijms-25-02109-f001:**
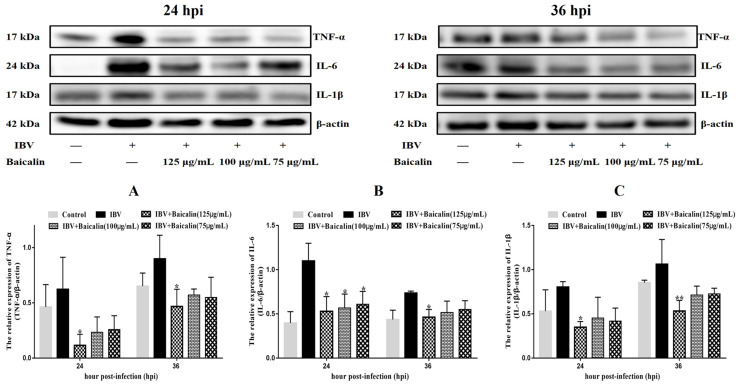
The effect of Bai treatment on levels of inflammatory cytokines. (**A**–**C**) represent the protein levels of TNF-α, IL-6 and IL-1β. Statistical significant are denoted by * *p* < 0.05, ** *p* < 0.01.

**Figure 2 ijms-25-02109-f002:**
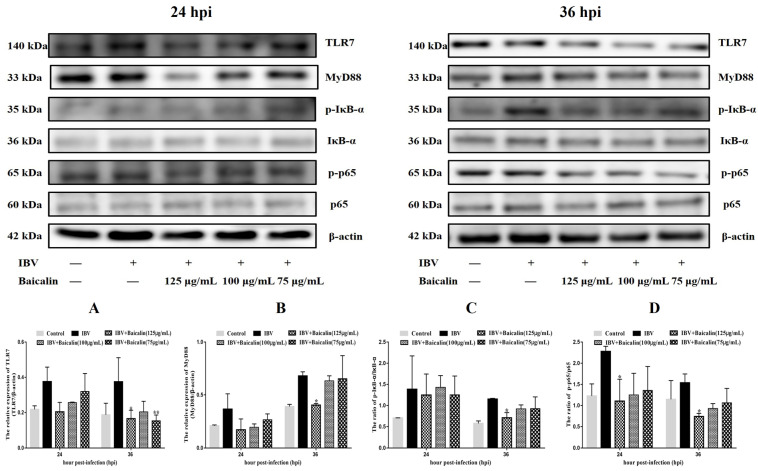
The protein levels of TLR7, MyD88, IκB-α and NF-κB p65 were presented from (**A**–**D**). Statistical significant are denoted by * *p* < 0.05, ** *p* < 0.01.

**Figure 3 ijms-25-02109-f003:**
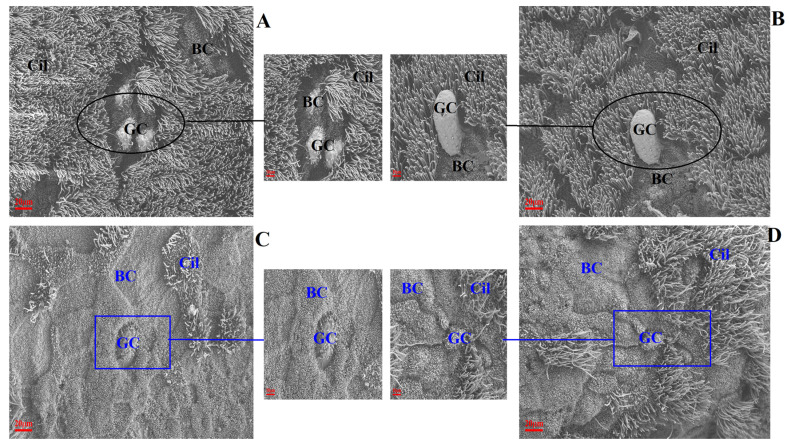
The effect of Bai on ultrastructural change in chicken trachea. (**A**–**D**) represent the ultrastructural morphology of chicken trachea in the control group, uninfected/Bai-treated group, untreated/IBV-infected group and Bai-treated/IBV-infected group at a magnification of 2500×, respectively. The black circle is used to indicate where cilia are abundant and intact in the uninfected control group (**A**) and Bai treatment group (**B**) at 5 dpi. The dark blue square indicates where cilia were damaged and shedding in the untreated/IBV-infected group (**C**) and the Bai treated/IBV-infected group (**D**) at 5 dpi. BC, CC, Cil and GC indicate brush cells, ciliated cells, cilia and goblet cells in turn.

**Figure 4 ijms-25-02109-f004:**
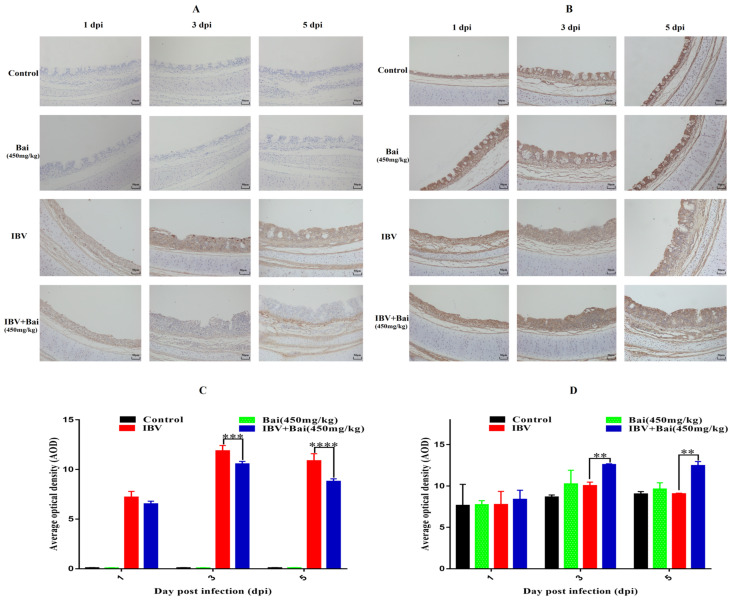
Effect of Bai on IBV N protein and G3BP1 in tracheal mucosa by IHC detection. (**A**,**B**) represent the localised expression of IBV N and G3BP1 on the tracheal surface at 1, 3, 5 dpi. (**C**,**D**) represent the statistical analysis of IBV N and G3BP1 expression on the tracheal surface at 1, 3, 5 dpi. Statistical significant are denoted by ** *p* < 0.01, *** *p* < 0.005, **** *p* < 0.001.

**Figure 5 ijms-25-02109-f005:**
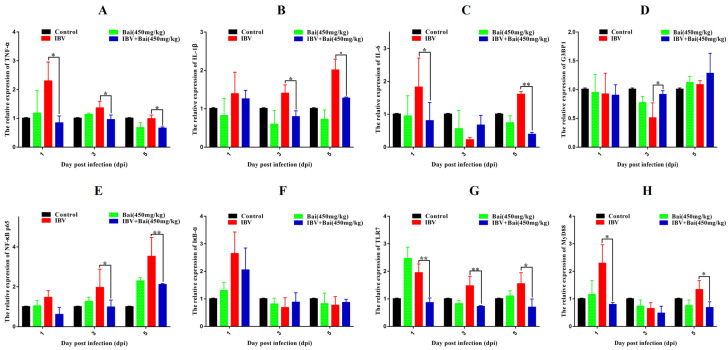
Bai inhibited the mRNA expression of inflammatory pathway and its downstream inflammatory factors in trachea. (**A**–**D**) represent the expression of TNF-α, IL-1β, IL-6 and G3BP1. (**E**–**H**) represent the expression of NF-κB, IκB-α, TLR7 and MyD88. Statistical significant are denoted by * *p* < 0.05, ** *p* < 0.01.

**Figure 6 ijms-25-02109-f006:**
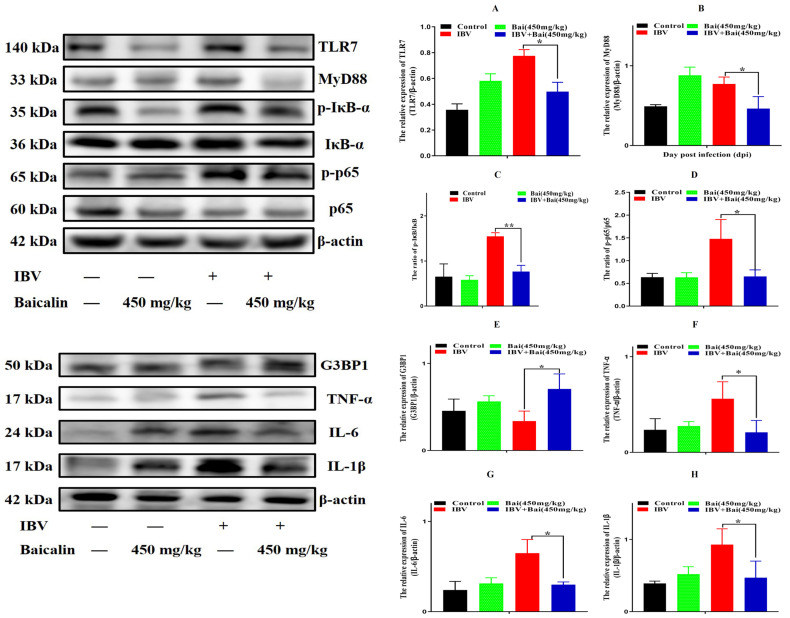
Bai inhibited the protein expression of inflammatory signal pathway and its downstream inflammatory factors in trachea. (**A**–**D**) represent the protein expression of TLR7, MyD88, IκB-α and NF-κB p65. (**E**–**H**) represent the protein expression of G3BP1, TNF-α, IL-6 and IL-1β. Statistical significant are denoted by * *p* < 0.05, ** *p* < 0.01.

**Figure 7 ijms-25-02109-f007:**
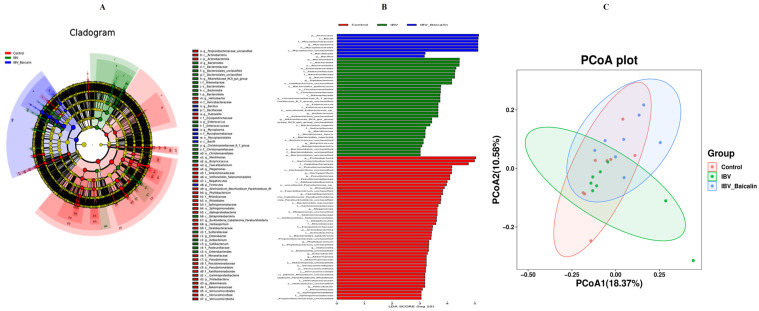
Effect of Bai on respiratory tract flora specific taxa. Linear discriminant analysis effect size prediction was used to identify the most differentially abundant bacteria in each group. (**A**,**B**) represents the expression of important flora in each group. In subtitle (**B**), dark blue represents uninfected and untreated group group, green represents untreated IBV-infected group, red represents Bai-treated/IBV-infected group. (**C**) represents the *beta* diversity of microbiota, which was evaluated by the PCoA based on the unweighted UniFrac distance.

**Figure 8 ijms-25-02109-f008:**
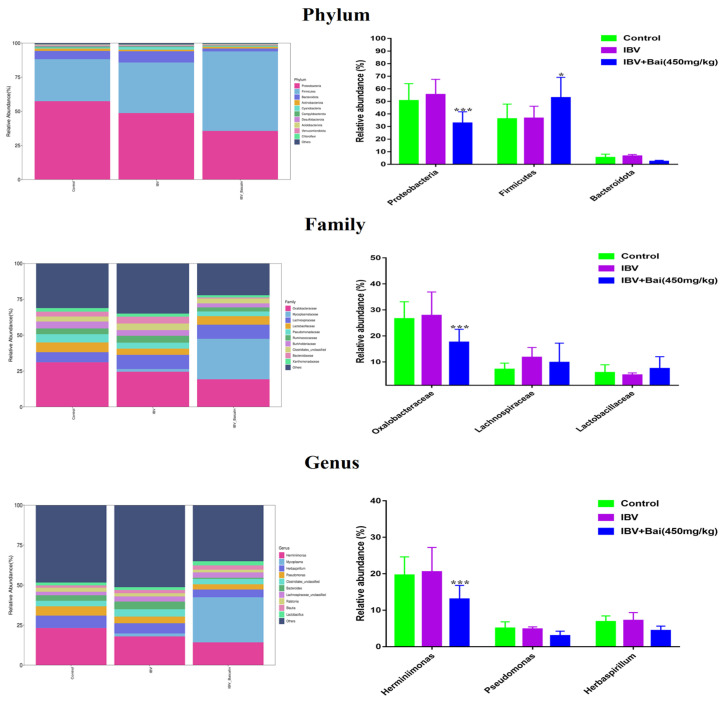
Effect of Bai on respiratory tract flora abundance. Statistical significant are denoted by * *p* < 0.05, *** *p* < 0.005.

**Figure 9 ijms-25-02109-f009:**
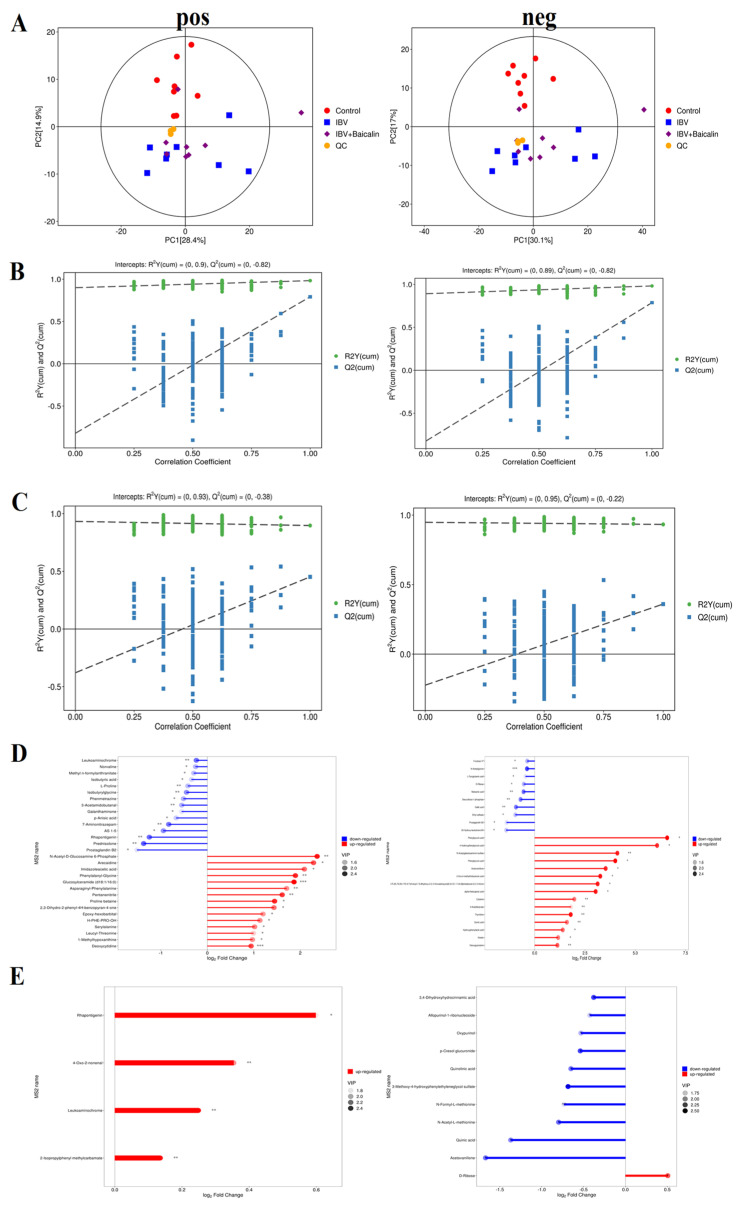
Screening and identification of the differential metabolites in the different groups. (**A**) represents the principal component analysis in both positive and negative ion mode. (**B**,**C**) represent partial least squares discrimination analysis in both positive and negative ion mode between different groups. (**D**,**E**) represent the matchstick diagram of both positive and negative ion modes between different groups. * *p* < 0.05, ** *p* < 0.01, *** *p* < 0.005.

**Figure 10 ijms-25-02109-f010:**
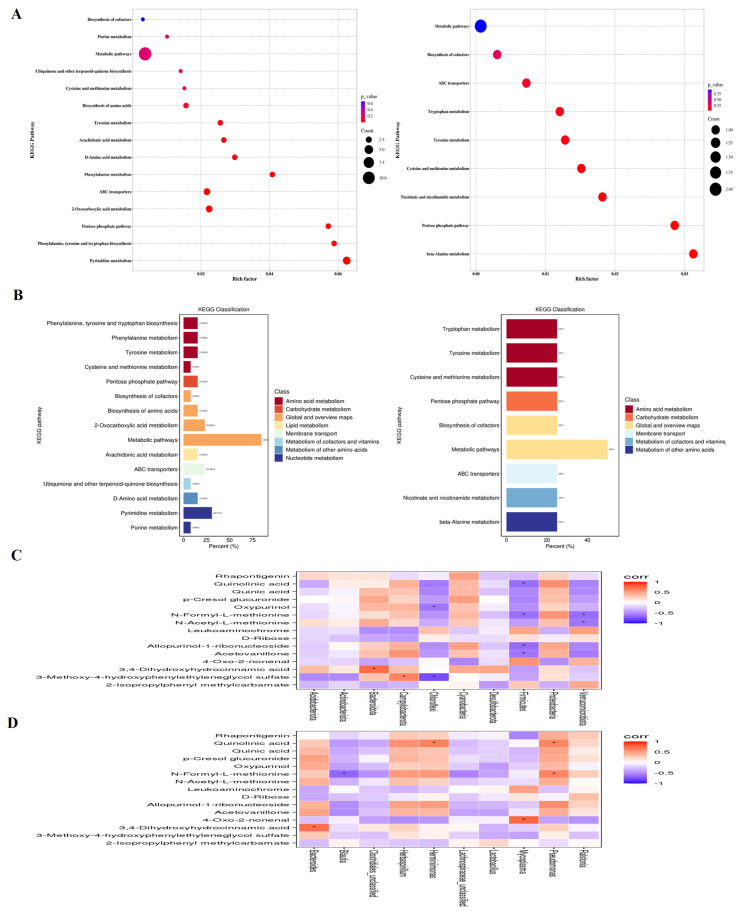
(**A**) represents metabolite pathways and classification annotation in both the positive and negative ion modes. (**B**) represents a KEGG enrichment bubble diagram of metabolic pathways in both positive and negative ion modes between untreated IBV-infected group and Bai-treated/IBV-infected group, as well as the correlation between differential metabolites and tracheal microbiota. (**C**,**D**) represent the correlation heat maps at both the genus and phylum levels, respectively. Statistical significant are denoted by * *p* < 0.05.

**Table 1 ijms-25-02109-t001:** The percentage body weight gain (%).

	Groups	Control	Bai	IBV	IBV + Bai
Days					
1 dpi		108.72 ± 13.37	114.69 ± 12.24	106.00 ± 11.29	115.46 ± 12.17
3 dpi		140.65 ± 8.63 ^a^	123.49 ± 8.41 ^a^	137.15 ± 9.61 ^ab^	131.93 ± 9.50 ^ab^
5 dpi		151.56 ± 12.19 ^a^	152.47 ± 6.64 ^a^	128.52 ± 7.28 ^b^	143.69 ± 7.88 ^a^

^a,b^ Different letters in the mean value of the same row indicate a significant difference (*p* < 0.05). The results are represented as the mean value ± SEM (n = 6).

**Table 2 ijms-25-02109-t002:** Details of antibodies used for IFA, IHC and WB.

Name	Dilution Ratio	kDa	Resource
IBV N	1:1000	50	Dayao Biotechnology (Hangzhou, China)
IL-6	1:1000	24	Cell Signaling Technology (Boston, MA, USA)
IL-1β	1:300	17	WanLei Biotechnology (Shenyang, China)
TNF-α	1:1000	17–26	SouthernBiotech (Birmingham, MD, USA)
TLR7	1:1000	140	Cell Signaling Technology (Boston, MA, USA)
MyD88	1:1000	33	Cell Signaling Technology (Boston, MA, USA)
NF-κB p65	1:250	65	Invitrogen (Carlsbad, CA, USA)
NF-κB p-p65	1:1000	65	Invitrogen (Carlsbad, CA, USA)
IκB-α	1:2000	36	Proteintech (Chicago, IL, USA)
p-IκB-α	1:500	35	Bioss (Beijing, China)
G3BP1 (WB)	1:1000	50	Abmart (Shanghai, China)
β-actin	1:1000	40	Abmart (Shanghai, China)
G3BP1 (IHC)	1:200	—	Abmart (Shanghai, China)
Goat Anti-Rabbit and Mouse IgG-HRP	1:5000	—	Abmart (Shanghai, China)
Goat anti-Mouse IgG AF488	1:300	—	Abmart (Shanghai, China)

**Table 3 ijms-25-02109-t003:** The primer sequences for RT-qPCR.

Gene	Primers (5′-3′)	Accession Number	Product Size (bp)
IBV N	F:GACGGAGGACCTGATGGTAA R: CCCTTCTTCTGCTGATCCTG	MK937830.1	206
IL-6	F:GTTCGCCTTTCAGACCTACCTG R:ATCGGGATTTATCACCATCTGC	NM_20468.1	130
IL-1β	F: CCTTCGACATCTTCGACATCAA R: AATGTTGAGCCTCACTTTCTGG	NM_204524.1	113
TNF-α	F: TGCTGTTCTATGACCGCC R:CTTTCAGAGCATCAACGCA	AY765397	219
TLR7	F:GCACACATTCAACTGGGGCAAAC R:TTCGGGGAACGGTAGTCAGAAGG	NM_001011688.3	115
MyD88	F: CGGAACTTTTCGATGCCTTTAT R: CACACACAACTTAAGCCGATAG	NM_001030962.5	107
NF-κB p65	F:ACCACCACCACCACAACACAATG R: GCGGCGTCGATGGTATCAAAGG	NM_001396038.1	116
IκB-α	F: GGATACCTGGCTGTTGTCGAATACC R: AAGTGTAGTGCTGTTCTCCCATTGC	NM_020529.3	89
G3BP1	F: AGGGTGAACAAGGTGATGTGGAAAC R: GCCATAGCCTGCAAGAGAAGAGC	NM_001006150.2	149
β-actin	F:CCCAAAGCCAACAGAGAGAA R: CCATCACCAGAGTCCATCAC	NM_205518	140

Abbreviation: IL-6, interleukin-6; IL-1β, interleukin-1β; TNF-α, tumour necrosis factor-alpha; TLR7, Toll-like receptor 7; MyD88, myeloid differentiation factor 88; NF-κB p65, nuclear factor kappa-B Rela; IκB-α, inhibitor of NF-κB; G3BP1, Ras GTPase-activating protein-binding protein 1; F, forward primer; R, reverse primer.

## Data Availability

All data generated or analysed during this study are included in this published article.

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
