# Peer review of "Baicalin Protects Broilers against Avian Coronavirus Infection via Regulating Respiratory Tract Microbiota and Amino Acid Metabolism"

_ijms, 2024, doi:10.3390/ijms25042109_

Round 1

Reviewer 1 Report

Comments and Suggestions for Authors

This manuscript presents interesting data, but the English is so poor that it is often hard to interpret the meaning of the data. There are also too many figures and tables; too much data to interpret. This manuscript would be better serves as two (or perhaps even three) separate manuscripts focusing on the different aspects of the study; in-vitro data, in-vivo data for the IBV portion (protection from challenge, histomorphometry, etc), and the in-vivo data regarding the microbiota fluctuation and the immune response.

Comments on the Quality of English Language

This manuscript needs to be extensively edited for English use by a native English speaker. There are too many grammatical and syntax errors to even begin to give examples. The English usage is so poor it makes the manuscript hard to understand and interpret.

Author Response

Dear Reviewer1,

Comment1: This manuscript presents interesting data, but the English is so poor that it is often hard to interpret the meaning of the data. There are also too many figures and tables; too much data to interpret. This manuscript would be better serves as two (or perhaps even three) separate manuscripts focusing on the different aspects of the study; in-vitro data, in-vivo data for the IBV portion (protection from challenge, histomorphometry, etc), and the in-vivo data regarding the microbiota fluctuation and the immune response.

Response: Thank you for your advices. We reduced the content of vitro data and focused on the vivo data. Meanwhile, we added some figures and all tables as supplementary materials in manuscript, and we simplified the content of the article to make it read more smoothly and fluently.

Comment 2: Comments on the Quality of English Language: This manuscript needs to be extensively edited for English use by a native English speaker. There are too many grammatical and syntax errors to even begin to give examples. The English usage is so poor it makes the manuscript hard to understand and interpret.

Response: We apologized for the poor language of our manuscript. We worked on the manuscript for a long time and the repeated addition and removal of sentences and sections obviously led to poor readability. We have now worked on both language and readability and have also involved native English speakers for language corrections. We really hope that the flow and language level have substantially improved. The revisions in this manuscript were highlighted with red.

Reviewer 2 Report

Comments and Suggestions for Authors

This paper on Baicalin is interesting as it included the lung microbiota. The in vitro effect on cells has already been published in Avian diseases, but not exactly with the same batches. Since the paper is quite large this part could be reduced in the present paper.

I have had one general problem, mostly with the figures. The letters are very small from fig 7 (a, b, c too small) to the phylogenetic study or diversity one. For the latter I could not read the name of bacteria... The choice of colour is not steady : in fig 6 the red and blue do not correspond to the same batches in A and C, and C and D, E. In fig 8 identify the groups more clearly.

The choice of immunological markers is justified. The choice of the different analyses is not: why use PCA, this particular discriminant analysis, diversity indices (Chao1, Shannon, and Simpson)? Spearman rho appears in text but not in figure; why Spearman? Because data are not gaussian? TCM is mentioned in discussion without explaining what it stands for. The clinical status of animals (0 to 2) should be explained: how was it coded. The analysis of weight gains should be rather with covariance analysis. How were the mortalities in the different groups?

The use of baicalin to protect against pulmonary disease is not new and there are several works on its use  for covid 19 and others. The main merit of the work is to show the modification of the microbiote and explain in vitro  how the protection works. A specific problem of the paper is the number of figures and their readibility that may hinder a clear evaluation.

For these reasons I propose major modifications.

Author Response

Dear Reviewer 2,

Comment 1: This paper on Baicalin is interesting as it included the lung microbiota. The in vitro effect on cells has already been published in Avian diseases, but not exactly with the same batches. Since the paper is quite large this could be reduced in the present paper.

Response: Thank you for your instructive advices. We have confirmed that Baicalin can inhibit IBV Beaudette strain replication in Vero cell and published on the avian pathology. Therefore, we reduced the part of vitro experiment in the manuscript according to your comment. The revisions in this manuscript were highlighted with red.

Comment 2: I have had one general problem, mostly with the figures. The letters are very small from fig 7 (a, b, c too small) to the phylogenetic study or diversity one. For the latter I could not read the name of bacteria... The choice of colour is not steady: in fig 6 the red and blue do not correspond to the same batches in A and C, and C and D, E. In fig 8 identify the groups more clearly.

Response: Thank you for your suggestions. We apologized for any inconvenience when you reviewed due to the small size of numbers and some Figures. According to your comments, we have divided Figure7 into supplementary Figure 5 and 6 in supplementary materials on order to read more smoothly. We also revised the size of Figure 13 and 14 in order to read the name of bacteria more clearly, and it was named as Figure 7 and 8 in revised manuscript. In Figure 6, we have modified the color of different groups on the bar chart to maintain consistency, it has added as Supplemental Figure 4 in supplementary materials. In addition, we revised the ABCD in Figure 8 with red to identify the group more clearly, and it was named as Figure 3 in revised manuscript.

Comment 3: The choice of immunological markers is justified. The choice of the different analyses is not: why use PCA, this particular discriminant analysis, diversity indices (Chao1, Shannon, and Simpson)? Spearman rho appears in text but not in figure; why Spearman? Because data are not gaussian? TCM is mentioned in discussion without explaining what it stands for. The clinical status of animals (0 to 2) should be explained: how was it coded. The analysis of weight gains should be rather with covariance analysis. How were the mortalities in the different groups?

Response: Thank you for your instructive suggestions. The respiratory microbiota analysis was completed by the BIOTREE Biological technology Co., Ltd., Shanghai, China, we reviewed some literatures and found that most researches used the Chao1, Shannon and Simpson index for analysis. Because Chao1 mainly estimates the number of species that included in the community. The Shannon index comes from information entropy, and the Shannon index is larger, the uncertainty is greater. The greater uncertainty means high diversity. The Simpson value range is between 0 and 1. The minimum Simpson value is 0 when there is only one species in the community, which is also our intuitive understanding of the minimum diversity. In this study, we used UniFrac’s weighted PCoA to analyze beta diversity. PCoA analysis aggregates samples together with high similarity in community structure, while samples with significant differences in community structure will be far apart. The result 3.8 was analyzing by using PCA analysis, which is a statistical method that converts a set of observed potentially correlated variables into linearly uncorrelated variables through orthogonal transformation. The color and shape of scatter points represent different groups, and the closer the distribution of sample points is, the more similar the types and contents of metabolites in the sample are. We inquired with the technical team in BIOTREE Biological technology Co., Ltd and confirmed that the correlation analysis of differential metabolites and microbiota composition was completed by using the Spearman program, and the results are presented in Figure 10CD. We re-checked the discussion in the manuscript, and found that the mention of TCM does not make much sense, so we have deleted the explanation of TCM. The clinical status of animals (0 to 2) should be explained as follows: 2 represents the significant clinical symptoms, 0 represents that there were no significant clinical symptoms, 1 was between 0 and 2. We have added it to the caption of Supplemental Figure 4. We also conducted an analysis of variance on body weight, but we think it was more clearly to present the changing trend by using a line chart. The broilers have no death cases throughout the entire experimental period, but it exhibited significant clinical symptoms during IBV infection.

Comment 4: The use of baicalin to protect against pulmonary disease is not new and there are several works on its use for covid 19 and others. The main merit of the work is to show the modification of the microbiote and explain in vitro how the protection works. A specific problem of the paper is the number of figures and their readibility that may hinder a clear evaluation. For these reasons I propose major modifications.

Response: Thank you for your instructive comments. We simplified the content of the article to make it read more smoothly and fluently, and added some figures and all tables as supplementary materials in manuscript. The revisions in this manuscript were highlighted with red.

Round 2

Reviewer 2 Report

Comments and Suggestions for Authors

I did not see  much improvement of the last figures wghere it is written in small letters and increasing the focus  shows lack of clarity. This could be improved.

I did not see the daily weight gains in any table (supplementary) or not published sections.

Minors:

l 51: present these abbreviations AIV, NDV, ALV, and MG infections

l 98: ad libitum in italics since it is latin

l 268: Chao 1 instead of chao 1

Author Response

Dear reviewer 2,

Comment 1: I did not see much improvement of the last figures wghere it is written in small letters and increasing the focus shows lack of clarity. This could be improved.

Response: Thank you for your instructive suggestion. According to your comment, we have revised the letter in last figure to make it clearer.

Comment 2: I did not see the daily weight gains in any table (supplementary) or not published sections.

Response: Thanks. We added the daily weight gains (The percent of initial body weight) as a Supplementary Table 3 in the manuscript.

Comment 3: l 51: present these abbreviations AIV, NDV, ALV, and MG infections; l 98: ad libitum in italics since it is latin; l 268: Chao 1 instead of chao 1.

Response: Thank you for your instructive comment. We have presented the full name of these abbreviations AIV, NDV, ALV with red highlight, and MG infections in Line 50 to 52, and revised “ad libitum” to italics with red highlight in Line 99, then revised “chao 1” to “Chao 1” in Supplementary Figure 7.
